# In Situ Formation of Nanoparticles on Carbon Nanofiber Surface Using Ceramic Intercalating Agents

Alex A. Burnstine-Townley [1,2], Sajia Afrin [1,2], Yuen Yee Li Sip [1,3], David Fox [1,2] and Lei Zhai [1,2,3,*]

1   NanoScience and Technology Center, University of Central Florida, Orlando, FL 32816, USA
2   Department of Chemistry, University of Central Florida, Orlando, FL 32816, USA
3   Department of Materials Science and Engineering, University of Central Florida, Orlando, FL 32816, USA
*   Correspondence: lzhai@ucf.edu

**Abstract:** Nickel silicide nanoparticles were prepared in situ on carbon nanofibers through pyrolysis of electrospun fibers containing poly(acrylonitrile) (PAN, carbon fiber precursor), silazane (SiCN ceramic precursor), and nickel chloride (nickel source). SiCN ceramics produced in carbon nanofibers during the pyrolysis expanded the graphitic interlayer spacing and facilitated the diffusion of metal atoms to the fiber surfaces, leading to the formation of nickel silicide nanoparticles at a reduced temperature. In addition, nickel silicide nanoparticles catalyzed an in situ formation of carbon nanotubes, with carbon sourced from the decomposition of silazane. The method introduces a simple route to produce carbon supported metal nanoparticles for catalysis and energy storage applications.

**Keywords:** nanoparticle; carbon fibers; ceramics; intergraphitic spacing; carbon nanotubes

## 1. Introduction

Metallic nanoparticles play an important role in modern technologies of sustainability and green energy harvesting. Large surface area and high catalytic activities grant their extensive application in pollutant decomposition [1], organic synthesis [2,3], and water splitting [4,5]. Metallic nanoparticles are often supported by carbon nanofibers with high electric conductivity as well as good thermal and chemical stability [6]. Strong adhesion of the nanoparticles to the conductive support is essential for efficient charge transport and catalytic efficiency [7]. Adhesion is typically achieved by extensive processing steps including hydrothermal treatments [8] or electrostatic interaction via surface functionalization [9].

To simplify the fabrication procedure, metal salt precursors have been added to the solutions of poly(acrylonitrile) (PAN, precursors of carbon fibers) to generate PAN fibers with metal ions followed by a pyrolysis to produce metal nanoparticles in situ. However, this method typically results in nanoparticles encapsulated within the carbon fiber, exhibiting unsatisfactory catalytic activity due to the limited exposure of nanoparticles [10–12]. The dense structures of the carbon fibers retard the diffusion of metal atoms from the fiber core to the surface. The problem has been remedied by the addition of sacrificial fillers such as poly(vinylpyrrolidone) to burn off and expose the inner surface area of the fibers [13] or the use of extreme processing conditions to overcome poor diffusion coefficients such as high temperatures and lengthy pyrolysis, as demonstrated by Bazargan et al., who reported pyrolysis conditions of 1400 °C for 20 h to produce nickel nanoparticles on carbon fiber surfaces [14].

In this work, nickel silicide nanoparticles were prepared in situ on carbon nanofibers through the pyrolysis of electrospun fibers containing poly(acrylonitrile) (PAN, carbon fiber precursor), silazane (SiCN ceramic precursor), and nickel chloride (nickel source). The diffusion of metal atoms in carbon fibers was improved by expanding graphitic layers using silazane [15,16] that readily incorporated into the in situ formed graphitic layers during the pyrolysis. After pyrolysis, the composite fibers contained both turbostratic graphite domains and amorphous SiCN domains as well as interphases of the two, where graphene

layers were spaced further apart by intercalated silicon [17]. This expansion was shown to improve the diffusion of nickel atoms and thus result in nanoparticles on the surface of the carbon nanofiber. Such an approach provides a general route to produce nanoparticles on conductive carbon fiber surfaces that have potential in various applications including catalysis, environmental remediation, and sensing.

## 2. Materials and Methods

A precursor solution was prepared by dissolving 8 wt% polyacrylonitrile (PAN) (Sigma-Aldrich) and 4 or 2 or 0 wt% nickel chloride hexahydrate (Fisher Scientific) in N,N-dimethylformamide (DMF) (Sigma-Aldrich). A total of 0-8 wt% silazane (Ceraset PSZ 20—EF: $SiNC_{1.4}H_{5.4}$ from Kion Corporation) was then added to the solution and heated at 85 °C for 3 h, then cooled to room temperature. Electrospinning was performed in a vertical configuration at a flow rate of 0.3 mL/h, source voltage 10 kV, and grounded drum collector rotating at 70 rpm, distance of 15 cm. The obtained fibers were stabilized in air at 260 °C for 2 h, then pyrolyzed under 50 standard cubic centimeters per minute (sccm) argon at a 2 °C/min ramp rate and held at 1000 °C for 2 h. The collected fibers were subjected to characterization without further purification. X-ray diffraction (XRD), X-ray photoelectron spectroscopy (XPS), scanning electron microscopy (SEM), energy dispersive X-ray (EDX), transmission electron microscopy (TEM) were conducted (detailed information is provided in the Supplementary Materials).

## 3. Results and Discussion

The carbon fibers with different concentrations of SiCN were produced by pyrolyzing PAN/silazane electrospun fibers with the PAN to silazane ratio ranging from 8 to 1. The XRD study of carbon fibers with various concentrations of SiCN dopants clearly showed a shift in the carbon diffraction peak (Figure 1a). With the increased PAN/silazane ratio in the preceramic fibers (i.e., pure PAN to 1:1), the carbon crystal plane (002) peak in the XRD of the fibers pyrolyzed at 1000 °C shifted to lower $2\theta$ values, from 24.6° (pure PAN) to 24.4° (8:1wt% PAN: silazane), 23.9° (5:1wt%), 23.2° (2:1wt%) and 21.9° (1:1wt%). The calculated interlayer distance increased from 3.62 Å for carbon fibers produced from pristine PAN to 4.06 Å for carbon fibers produced from 1:1wt% PAN:silazane (Figure 1b), suggesting that SiCN dopants expanded the layer distance of graphitic domains.

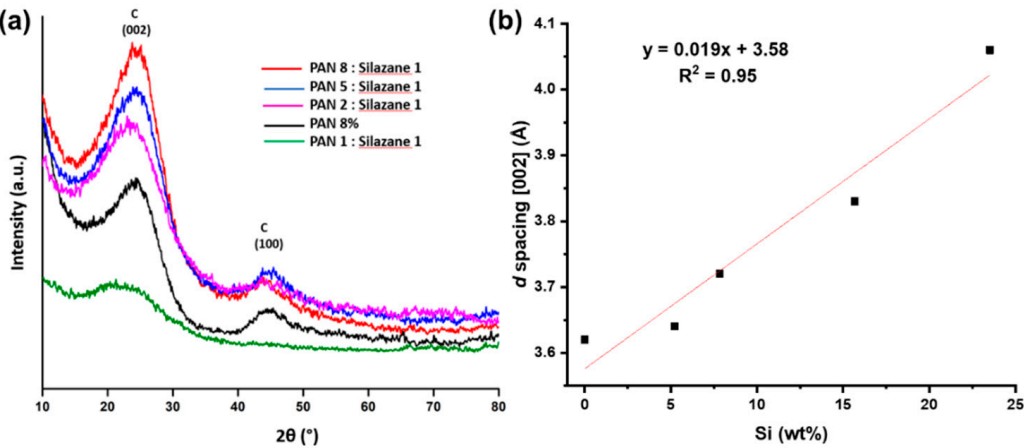

**Figure 1.** (**a**) XRD of carbon fibers obtained at 1000 °C with varying PAN:silazane ratios; (**b**) plot of (002) spacing vs. Si wt%, data from (**a**).

To examine the effect of the expanded graphite lattice on nanoparticle formation, nickel chloride was added to the precursor solution. Electrospinning an 8:1:4 PAN:silazane:$NiCl_2$ (by wt%) solution generated uniform, cylindrical nanometer-scale fibers (Figure S1). Pyrolysis of the fibers at 1000 °C for 2 h created nanoparticles on the fiber surface during the polymer to ceramic transformation (Figure 2a–c). The fibers were observed to be

350–400 nm in diameter. The morphology of nickel nanoparticles on the fibers was characterized to assess the diffusive mobility of atoms within the fiber. The enhanced atomic diffusion, brought on by the expanded graphite layer, allowed the nickel atoms to reach the surface more rapidly. Specifically, nanoparticles were formed in only 2 h at 1000 °C compared to the previously mentioned literature reporting pyrolyzing at 1400 °C for 20 h exclusive of silazane [14]. This major change in diffusion characteristics signifies the pivotal role of SiCN dopants to afford greater microporosity among the graphite layers. Furthermore, the nanoparticles were formed on the surface during carbonization, ensuring strong adhesion of the nanoparticles to the carbon fiber surface.

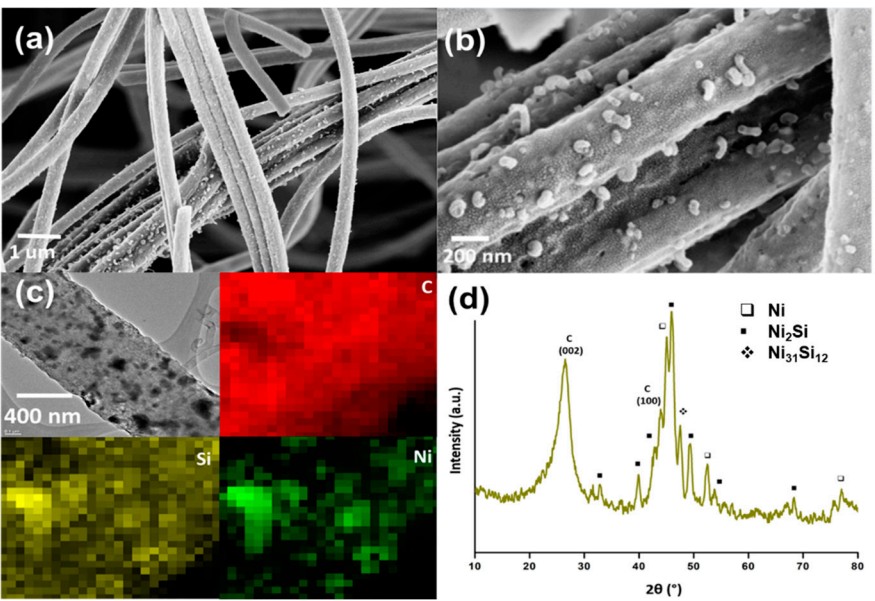

**Figure 2.** (**a,b**) SEM images, (**c**) TEM image, and EDX scans, red: carbon, yellow: silicon, green: nickel (**d**) XRD diffractogram of 8:1:4 PAN:silazane:NiCl$_2$ fibers pyrolyzed at 1000 °C.

The incorporation of silazane with the nickel chloride changed the reactant stoichiometry to form different species of nanoparticle products. TEM images with EDX mapping (Figure 2c) showed the formation of intermetallic nanoparticles composed of nickel and silicon. The formation of nickel silicide nanoparticles on the fiber surface confirmed the mobility of both the silicon and nickel atoms during pyrolysis. The XRD diffractogram (Figure 2d) showed the formation of various phases including metallic nickel, Ni$_2$Si, and Ni$_{31}$Si$_{12}$ [18,19]. XPS further confirms the presence of these species (Figure S2), with peaks corresponding to the Si–Ni and C–Si bonds [20–22].

Control experiments sought to produce in situ nickel nanoparticles on the carbon fiber surface without silazane by employing an 8:4 PAN:NiCl$_2$ (by wt%) precursor solution. Upon pyrolysis at 1000 °C, XRD did show the formation of metallic nickel nanoparticles (Figure S3), but none were observed on the surface of the fibers in the SEM images (Figure S4). We believe that inadequate lattice spacing restricted the diffusion of metal atoms to reach the fiber surface. Upon the exclusion of SiCN dopants, the graphitic (002) peak shifted from 26.7° (3.33 Å) (Figure 2d) to 26.5° (3.36 Å), demonstrating the lattice contraction in the absence of SiCN dopants.

Examining the pyrolysis temperature revealed the thermodynamic dependence of nanoparticle formation and the accompanying diffusion characteristics. In contrast to the formation of a crystalline carbon fiber composite support with Ni nanoparticles on the fiber surfaces upon pyrolysis at 1000 °C, the pyrolysis of 8:1:4 (PAN:silazane:NiCl$_2$) fibers at 600 °C did not create nanoparticles (Figure 3a). However, small nanoparticles first (10-15 nm) appeared at 700 °C and coalesced into larger nanoparticles on the fiber surface at higher temperatures (Figure 3b–d and Figure S5). This observation was different from a previous report showing nickel nanoparticles produced at 600 °C with a nickel

phenanthroline complex as the carbon source [19]. The difference was caused by the generation of reducing agents (e.g., $H_2$) through the thermal decomposition of the additives (phenanthroline vs. silazane) at different temperatures. Hydrogen generated through the decomposition of silazane occurred above 600 °C [23]. Increasing the temperature to 800 °C and 900 °C generated nanoparticles of size 25–35 nm and 50–70 nm, respectively (Figure 3c,d). The increasing particle size indicates that metal diffusion becomes more rapid as the temperature increases due to the expansion of the intergraphitic spacing, in stark contrast to the silazane-free formulation that resulted in trapped, subsurface nanoparticles.

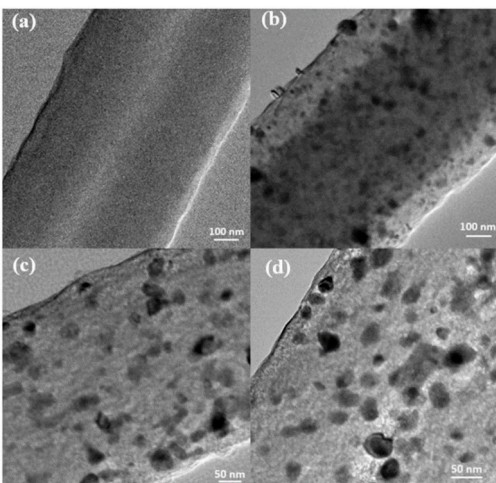

**Figure 3.** TEM images of 8:1:4 (PAN:silazane:$NiCl_2$) fiber samples pyrolyzed at (**a**) 600 °C, (**b**) 700 °C, (**c**) 800 °C, (**d**) 900 °C.

It was interesting to observe carbon nanotube (CNT) formation on the nanoparticles during pyrolysis (Figure 4). Nickel silicide has been shown to catalyze the growth of CNTs at 600–700 °C [24], paired with the carbon sourced from decomposing silazane (i.e., $H_2$ and $CH_4$). The nanoparticles remained on the nanofiber surfaces during the growth of the CNTs, suggesting a strong adhesion of the nanoparticle to fiber surfaces. Although CNT production was not the primary focus of this study, the observation demonstrated the potential of carefully selected catalysts and reagents for in situ production methods of additional nanomaterials beyond the PDC composites.

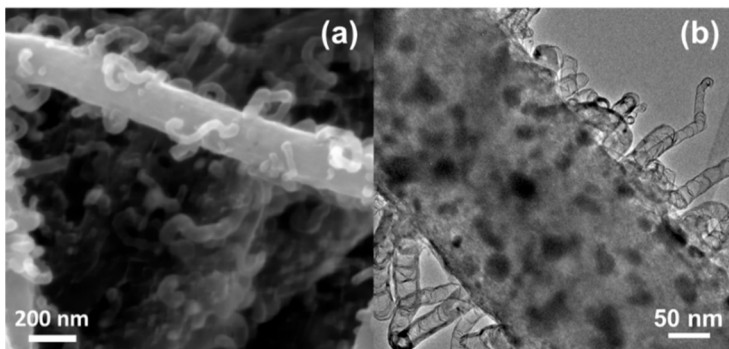

**Figure 4.** (**a**) SEM and (**b**) TEM images of the carbon nanotubes formed on 8:1:4 (PAN:silazane:$NiCl_2$) fibers pyrolyzed at 1000 °C.

## 4. Conclusions

In summary, an effective approach to produce nickel and nickel silicide nanoparticles on carbon fiber surfaces was developed by electrospinning a solution of PAN, silazane, and nickel chloride followed by pyrolysis. The expansion of the carbon intergraphitic spacing by SiCN dopants in the carbon fibers improved the diffusion of metal atoms to the fiber surface, enabling the in situ generation of metallic nanoparticles on the fiber

surface during pyrolysis. In addition, these nanoparticles can catalyze the formation of CNTs with a carbon source provided through the decomposition of silazane. The method grants a simple approach to produce various nanoparticles on carbon fibers for numerous catalytic applications.

**Supplementary Materials:** The following supporting information can be downloaded at: https://www.mdpi.com/article/10.3390/jcs6100303/s1. Figure S1: SEM image of 8:1:4 PAN:silazane:NiCl$_2$, stabilized at 260°C. Figure S2: XPS spectra of 8:1:4 PAN:silazane:NiCl$_2$, pyrolyzed at 1000°C. (a) C 1s, (b) Si 2p, (c) Ni 2p, (d) N 1s, (e) O 1s. Figure S3: XRD pattern of 8:4 PAN:NiCl$_2$, pyrolyzed at 1000°C. Figure S4: SEM images of 8:4 PAN:NiCl$_2$, pyrolyzed at 1000°C. Figure S5: SEM images of 8:1:4 PAN:silazane:NiCl$_2$, pyrolyzed at temperature (a) 600°C, (b) 700°C, (c) 800°C, (d) 900°C.

**Author Contributions:** Conceptualization, writing—original draft preparation, A.A.B.-T. and S.A.; Methodology A.A.B.-T. and S.A.; Investigation, A.A.B.-T., S.A., Y.Y.L.S., and D.F.; Supervision, writing—review and editing, L.Z. All authors have read and agreed to the published version of the manuscript.

**Funding:** This research was funded by the University of Central Florida Director Startup.

**Data Availability Statement:** Data available upon request.

**Acknowledgments:** The authors acknowledge the UCF MCF-AMPAC facility, MSE, and NSTC.

**Conflicts of Interest:** The authors declare no conflict of interest.

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
