# Peer review of "In Situ Formation of Nanoparticles on Carbon Nanofiber Surface Using Ceramic Intercalating Agents"

_jcs, doi:10.3390/jcs6100303_

Round 1

Reviewer 1 Report

I found several misspelling or confusing words such as siliazane (line 44) and silizane (line 70).

2, I could not find any color difference for carbon, silicon, nickel in Fig. 2.

 I could not agree with the author on the suggestion in line 128. In Fig. 3 and s5, any distinctive size difference of nanoparticles was shown. 

Spacing word was needed for line 135. 

(respectively(Figure..)..  -> respectively (Figure..)..

Author Response

I found several misspelling or confusing words such as siliazane (line 44) and silizane (line 70).

The typos are fixed. 

 I could not find any color difference for carbon, silicon, nickel in Fig. 2.

 I could not agree with the author on the suggestion in line 128. In Fig. 3 and s5, any distinctive size difference of nanoparticles was shown.

We believe that the increase of particles size was caused by the diffusion of metal atoms similar to Ostwald ripening. 

Spacing word was needed for line 135.

(respectively(Figure..)..  -> respectively (Figure..)..

The error is fixed. 

Reviewer 2 Report

Alex A. Burnstine-Townley et al demonstrated In situ formation of nanoparticles on carbon nanofiber surface using silazane based intercalating agents. Their method introduces a relatively simple route to produce carbon supported metal nanoparticles. I would like to recommend this article to be accepted in this journal with a minor comment:

·       The size and quality of Figure S2 is small, so data is not visible properly. So, Increase the size and quality of the Figures S2

Author Response

The size and quality of Figure S2 is small, so data is not visible properly. So, Increase the size and quality of the Figures S2

The font size and image size were increased to make them more visible. 

Reviewer 3 Report

In this work, a route for obtaining Ni2Si / Ni nanoparticles decorated carbon nanofibers has been developed. SiCN precursor mixed with PAN in the electrospinning solution induces the formation of turbostratic graphite with increased interlayer space, allowing the diffusion of metal to the surface of the nanofibers. The work is relevant in the field of hybrid nanomaterials with applications in catalysis and energy. It is well written and only a few points need some clarification. Minor revision is recommended.

1. Please include a brief comment on specific applications of the fabricated material in the Introduction section.

2. Indicate the average diameter of the produced fibers.

3. Check references match journal´s format.

Author Response

Please include a brief comment on specific applications of the fabricated material in the Introduction section.

A comment was added at the end of the introduction. 

Indicate the average diameter of the produced fibers.

The average size of fibers (350-400 nm) was added to the manuscript. 

Check references match journal´s format.

The format was fixed.